# Clinical Profiles in Multiple Sclerosis: Cognitive Reserve and Motor Impairment along Disease Duration

**DOI:** 10.3390/bs13090708

**Published:** 2023-08-25

**Authors:** Sabrina Maffezzini, Veronica Pucci, Alice Riccardi, Sonia Montemurro, Marco Puthenparampil, Paola Perini, Francesca Rinaldi, Paolo Gallo, Giorgio Arcara, Sara Mondini

**Affiliations:** 1Multiple Sclerosis Centre, University-Hospital of Padua, 35128 Padova, Italy; sabrina.maffezzini@studenti.unipd.it (S.M.); alicericcardi13@gmail.com (A.R.); marco.puthenparampil@unipd.it (M.P.); dr.paola.perini@gmail.com (P.P.); dr.francesca.rinaldi@gmail.com (F.R.); paolo.gallo@unipd.it (P.G.); 2Department of Philosophy, Sociology, Education and Applied Psychology, University of Padua, 35128 Padova, Italy; veronica.pucci@phd.unipd.it; 3Human Inspired Technology Centre (HIT), University of Padua, 35122 Padova, Italy; 4IRCCS San Camillo Hospital, 30126 Venice, Italy; sonia.montemurro@hsancamillo.it (S.M.); giorgio.arcara@hsancamillo.it (G.A.); 5Department of Neuroscience, University of Padua, 35128 Padova, Italy

**Keywords:** multiple sclerosis, cognitive impairment, motor impairment, disease duration, cognitive reserve, generalised additive models

## Abstract

(i) Background: Cognitive impairment in people with multiple sclerosis (MS) has been studied in relation to certain clinical variables (e.g., motor disability and disease duration) and lifestyle factors such as cognitive reserve (CR). However, only very few studies have considered the interaction of clinical variables and cognitive reserve in preserving the integrity of the neuropsychological profile. In this paper, we hypothesised that a higher level of CR might predict good cognitive efficiency by modulating the clinical outcome of the disease. (ii) Methods: A sample of 100 participants with MS (age range 30–74), was recruited and assessed remotely with a questionnaire to measure CR and a cognitive screening test. Data were analysed through generalized additive models. (iii) Results: We found that the model analysing the interaction between CR and disease duration, and between CR and motor disability, was able to explain a significant percentage of cognitive performance. In particular, higher levels of CR predicted a better cognitive performance despite a long disease duration, unless the motor disability was severe. (iv) Conclusion: This study highlights the crucial role of CR in modulating cognitive efficiency in people with MS.

## 1. Introduction

Multiple sclerosis (MS) is a chronic, inflammatory, degenerative disease affecting both grey and white matter of the central nervous system [1].

Cognitive impairment occurs in up to 70% of individuals with MS [2,3,4], who typically show affected cognitive functioning in the domains of sustained attention, information processing speed, memory, and executive functions [5,6,7] among other changes in their quality of life. For this reason, the assessment of the neuropsychological status of people with MS is important in diagnosis and during the monitoring of the progression of the disease [2].

Although cognitive impairment is one of the most frequent symptoms of MS, it is not clear why some patients do not show such dysfunctions [8,9], whereas others show a decline. These latter individuals are usually older [10] and have a longer history of disease (i.e., from here onwards reported as “disease duration”) [11].

Other studies have focused on the description of cognition in individuals with MS based on their motor disability, measured on the Expanded Disability Status Scale, i.e., the EDSS [12], which is a routinely used method for quantifying motor disability in MS and monitoring changes over time. Recent longitudinal studies have indeed shown that higher motor disability is correlated with lower cognitive functioning [13,14], even if the relationship between motor and cognitive impairment is still controversial [15].

In recent years, cognitive reserve (CR) has been considered as a construct able to explain the possible variance of cognitive impairment in individuals with MS [16,17], similar to its function in healthy older populations at risk of dementia [18]. CR allows for sustained cognitive performance when individuals have to face age-related changes, brain damage, or neurological disease [19,20,21]). CR derives from lifetime experiences characterised by intellectual activity and environmental factors, all things that allow individuals to maintain a good cognitive efficiency [22] and independence in daily living. Several proxies are used to quantify CR: education, occupational attainment (i.e., work complexity and responsibility required), engagement in leisure and social activities (such as playing a musical instrument, reading books and newspapers, gardening, traveling, or going to the cinema and theatre), IQ level (measured with specific and complex scales), and vocabulary size [23,24]. One of the most used tools to quantify CR, in a standardised way, is the cognitive reserve index questionnaire (CRIq) [25].

A certain amount of research has examined the specific role of CR in modulating cognitive functioning in individuals with MS [26,27]. In general, the results of such studies have shown that high levels of CR may be associated with lower cognitive dysfunction [28,29,30]. Instead, together with old age, low levels of CR are one of the most significant predictors of cognitive decline in individuals with MS [31,32].

However, relatively few studies have focused on the precise role of CR in the cognition of individuals with MS while also considering clinical indices such as the disease duration, age of onset, and level of motor disability. Rimkus and colleagues [33] suggested that the protective effect of CR is strongest in the early years after the disease onset when there is a reduced effect of inflammatory activity on cognitive abilities. A meta-analysis [28] has highlighted the only effect of CR on cognitive performance and not on other clinical variables such as disease severity, disease duration, motor disability (measured with EDSS), and type of MS. Furthermore, Artemiadis et al. [34] underlined that in individuals with the same level of motor disability, those with a higher CR showed a better cognitive performance in a series of neuropsychological tasks.

These findings, however, are controversial and further investigation is needed. Indeed, CR was quantified through a heterogeneous pool of proxies and, in most cases, without a composite index. Tremblay et al.’s [35] investigation is one of the very few previous studies which considered a composite measure of CR using the standardised and reliable cognitive reserve index questionnaire (CRIq) [25]. The authors found that CR moderated the relationship between EDSS and cognition. However, this investigation did not take into account the relevant clinical factor of disease duration.

The purpose of the present investigation is to verify whether a higher level of CR may interact with disease duration and motor disability in predicting cognitive outcomes. In particular, we hypothesised that, despite disease duration and motor impairment, a higher level of CR (quantified with the CRIq) may allow individuals with MS to maintain a good cognitive efficiency. Due to the multiple factors considered in our hypothesis, we analysed the data through generalised additive models (GAMs) [36], a statistical approach which allowed us to verify the complex interaction among multiple continuous variables.

## 2. Materials and Methods

### 2.1. Participants

In this cross-sectional study, 100 individuals with MS were enrolled: 88 with relapsing–remitting MS and 12 with progressive MS (according to the recent guidelines by Lublin et al., 2015) [37]. Participants’ (72 F and 28 M, F: M = 2.6:1) mean age was 50.9 ± 9.25 (range 30–74), mean education was 12.5 ± 3.53 years (range 5–21), disease duration ranged from 1 to 37 years (mean = 14.82 ± 9.28), and motor disability, measured through the EDSS, ranged from 1.0 to 8.0 (mean = 3±1.89). Most participants were treated with first- or second-line drugs, whereas 7% of them had no pharmacological therapy for a number of reasons (e.g., personal choice, no availability of appropriate drugs for the typology of MS).

The inclusion criteria were: (i) age >18, (ii) no history/evidence of psychiatric or neurological disorders other than multiple sclerosis, (iii) no history of alcohol or drug abuse, (iv) no hearing difficulties, (v) Italian mother tongue, and (vi) individuals without a recent relapse of the disease.

### 2.2. Procedure

Participants were recruited between January and July 2021 at the Multiple Sclerosis Centre of the University Hospital of Padua, where they arrived for a first diagnosis or for clinical follow-ups. When a participant suitable for this study was at the Multiple Sclerosis Centre, the EDSS was administered by a trained neurologist. An expert psychologist scheduled the telephonic cognitive assessment no more than one week later. The entire procedure of this study was conducted remotely, by telephone. Each participant was informed about the research purpose and their consent was acquired via audio-recording. The examiner ensured that the room was quiet and distraction-free and that the telephone connection was stable.

All participants first answered the cognitive reserve index questionnaire (CRIq) [25] and then underwent the cognitive assessment by the Tele-Global Examination of Mental State, version A (Tele-GEMS) [38]. The whole administration lasted about 20 min.

The study was approved by the Ethical Committee of the Hospital of Padua (Prot. N 19669), and it was conducted in accordance with the Declaration of Helsinki.

### 2.3. Assessment and Materials

For each participant, clinical variables, cognitive reserve, and neuropsychological measures were accurately collected.

Clinical variables: included (i) disease duration (number of years) registered in the clinical report and personal anamnesis; (ii) motor disability, assessed through the EDSS [12] administered by trained neurologists. The other two tools were administered by a trained psychologist.

Cognitive reserve was measured with the cognitive reserve index questionnaire (CRIq) [25], which is a semi-structured interview to quantify a person’s CR considering their education (CRI-Education), working activity (CRI-WorkingActivity), and leisure time activities (CRI-LeisureTime) carried out during adulthood. The administration of CRIq provides the global index of cognitive reserve (CRI). A full description of the 20 items included in the CRIq and instructions for its administration are freely available at https://www.cognitivereserveindex.org/ (accessed on 27 June 2023).

Neuropsychological assessment was aimed to evaluate cognitive functioning through the Tele-Global Examination of Mental State (Tele-GEMS) [38], a new cognitive screening tool that is administered remotely using a telephone and consists of ten tasks investigating different cognitive domains: orientation, memory, working memory, spatial representation, language, auditory attention, verbal fluency, and pragmatic abilities. It provides an index of global cognitive functioning, which reflects the balanced contribution of each task to a total score that ranges from 0 to 100. The administration lasts about 10 min. This tool has good psychometric properties and it has been proven to be equivalent to its in-person version (i.e., GEMS). Instructions, score sheets, and cut-offs according to age, education, and CR are available at https://osf.io/r3ta5/ (accessed on 27 June 2023).

### 2.4. Statistical Analysis

All statistical analyses were performed with R software [39]. First, an independent sample *t*-test was carried out to analyse possible differences according to sex in order to verify whether to include this variable in the subsequent analyses. Descriptive correlation analyses among our predictors were calculated through Pearson’s *r*. Generalised additive models (GAMs) [36] of increasing complexity were used to investigate the interaction between clinical variables (i.e., EDSS and disease duration) and cognitive reserve (CRI) on the one side, and cognitive functioning (Tele-GEMS) on the other.

The main advantage of GAMs over the more traditionally used GLMs is their ability to efficiently model, based on smooth functions, nonlinear relationships between the predictors and the dependent variable. Although this is also possible with GLMs, in GLMs nonlinearity must be specified explicitly a priori. In contrast, the modelling procedure in a GAM uses a bottom-up approach to estimate whether a nonlinear relationship improves the fit. In GAMs, the relationship between the predictor and the expected value of the dependent variable is modelled through a smooth function, which in turn may follow any exponential family distribution or simply have a known mean–variance relationship, allowing the use of a quasi-likelihood approach.

In our analyses, the models were built from simpler to more complex in order to examine the best pattern of variables that would fit our data. The goodness of fit of the models was assessed with the Akaike Information Criterion (AICc), and GCV was used in order to check for any possible overfitting (GCV) [40,41]. We also calculated the Akaike Information Criterion weights (AICw) which is the probability of a model being the best among a given set of models and, as such, can be used for model selection. The best model was chosen according to the AICw (R^2^ and AIC are also reported) and a series of diagnostic tests were carried out to check the appropriateness of the model characteristics. We also checked the quality of the GAM models by visual inspection of the residuals.

## 3. Results

Descriptive statistics of our variables (age, education, disease duration, EDSS, CRI, and Tele-GEMS) are reported in Table 1.

Our sample did not show any difference between males and females on Tele-GEMS performance (*t*(98) = −0.79, *p* = 0.43); thus, sex was not included as a predictor in the GAM models.

Correlational analyses showed that Tele-GEMS correlated positively with CRI (*r* = 0.50, *p* < 0.001) and negatively with age (*r* = −0.29, *p* = 0.003), EDSS (*r* = −0.44, *p* < 0.001), and disease duration (*r* = −0.23, *p* = 0.02). Correlations among all of our variables of interest (Tele-GEMS, CRI, age, EDSS, and disease duration) are reported in Table 2.

Generalised additive models (GAMs) analyses underlined that the best model was the one assessing the possible influence between CRI and EDSS, correcting for age and disease duration (Model 4). In particular, Model 4 had the highest R^2^ (R^2^ = 0.458 and 50.9% deviance explained) and the lowest AIC. We chose Model 4 as it was the best according to its AICw (See Table 3). A series of diagnostic tests (R function ‘gam.check’) showed appropriate characteristics for the tensors referring to the interaction effects (for CRI*Disease Duration: k′ = 4, *p* = 0.78; for CRI*EDSS: k′ = 4, *p* = 0.99).

A visual inspection of Model 4 related to the interaction between CRI and disease duration is shown in Figure 1, and a similar inspection related to the interaction between CRI and EDSS is shown in Figure 2. These two figures give additional information on the interaction between CRI and EDSS and disease duration in predicting cognitive performance (i.e., Tele-GEMS). Tele-GEMS scores are represented with different colours: the darker shades of blue in the contour plot represent lower scores, while the darker shades of yellow represent higher scores. Figure 1 shows the interaction between CRI and disease duration, highlighting that high levels of CRI (on the *x*-axis) are associated with a better Tele-GEMS (on the *y*-axis) even when the disease duration (on the *z*-axis) is very high. However, a negative effect of disease duration is found on the Tele-GEMS in persons with a low CR. Figure 2 shows the interaction between CRI and EDSS, indicating that high levels of CRI (on the *x*-axis) correspond to a higher Tele-GEMS (on the *y*-axis), but that such a positive effect of CRI is significantly attenuated when the EDSS (on the *z*-axis) is very high (i.e., severe motor disability). This is clear in Figure 2 (right side) in correspondence with the dark blue in the upper right-hand-side corner. In the case of a low CR, instead, the Tele-GEMS is low at any level of EDSS.

## 4. Discussion

The present study aimed to investigate the impact of important factors such as CR, disease duration, and motor disability on the cognitive efficiency of people with multiple sclerosis. It is well known that high levels of CR are associated with better cognitive performance in both healthy [42] and pathological people, such as those who have dementia [43] or MS [30]. Our data, obtained with a sample of 100 people with MS, are in line with what has already been reported in the literature [11,14]: cognitive efficiency is negatively correlated with disease duration and with motor disability but positively correlated with CR. The added value of the present study is the methodological approach adopted. Indeed, the use of GAMs [36] allowed us to consider the potentially complex interaction between different clinical variables and socio-demographic variables, at the same time disentangling their respective effects: controlling for age, the data showed that motor disability and disease duration in interaction with CR were significant predictors of cognitive performance. These results confirm the positive effect of CR in predicting a better cognitive profile regardless of disease duration and motor disability but not in the case of severe motor impairment (i.e., EDSS = 6–7, corresponding to needing bilateral assistance to walk 100 m). As MS is a degenerative disorder, our findings (i.e., an extended positive effect of CR throughout the disease) are crucial, since they suggest that patients with a high CR may maintain a suitable quality of life for longer. However, at a high level of motor impairment, CR is no longer protective and, at this point in the disease, a cognitive decline appears inevitable. In line with our results, Tremblay et al.’s (2023) [35] research highlighted the modulating effect of CR on motor disability and underlined its effect on specific cognitive domains (i.e., visuospatial memory and processing speed); however, they had not considered disease duration, which we found to be an important clinical variable that we cannot neglect.

This positive effect of CR on cognition could be explained in terms of the corresponding brain reserve (including the richness of networks and dendritic spines, synaptic density, compensation mechanisms, and brain plasticity). In other words, CR is strictly correlated to the variations of the cerebral capacity according to the environment and the processes of plasticity. In particular, two processes are involved, the first is the one used by the healthy brain during the execution of various intellectual tasks, which implies a more efficient use of neural networks. The second, the compensatory mechanism, is activated in the event of brain injury and the neural networks utilised for cognitive functions differ from those that are observed in individuals who have a well-preserved neurological function. These brain/cognitive mechanisms support and maintain a better cognitive efficiency [19].

The present cross-sectional investigation, however, is not exempt from limitations. First of all, MS is a progressive disorder and any possible effect of CR over time should be further investigated through longitudinal observations, in which the clinical changes in MS patients would be systematically monitored. Secondly, neuroimaging data were not available for this study. However, if on the one hand the use of such data would have allowed us to better characterise brain deterioration, on the other, GAMs provided a solid pool of information accounting for the complex interaction among different clinical variables.

However, other important variables were possibly at play in affecting cognitive efficiency in our sample such as perceived fatigue, type of medication, mood deflection, and psychological traits. These confounding factors have not been considered in the present study, and thus further investigation will be necessary to deepen and examine the interaction between CR and cognitive MS symptoms.

In addition, the cognitive assessment was conducted remotely, and this may represent a possible limitation in the reliability of the cognitive measures. However, we adopted a standardised cognitive screening test (Tele-GEMS), administered by an expert psychologist. Tele-GEMS was specifically designed for the telephone-based assessment, and it has been proven to be equivalent to in-person cognitive screenings [38].

## 5. Conclusions

In conclusion, this study reveals that accounting for the interplay between CR and disease duration, as well as between CR and motor disability, may explain a significant degree of variance in cognitive performance. Our results, which we hope will be of use to clinicians and researchers, indicate that high levels of CR are related to an improved cognitive performance even in prolonged disease duration, although severe motor disability may, at some point, attenuate this “protective effect”.

## Figures and Tables

**Figure 1 behavsci-13-00708-f001:**
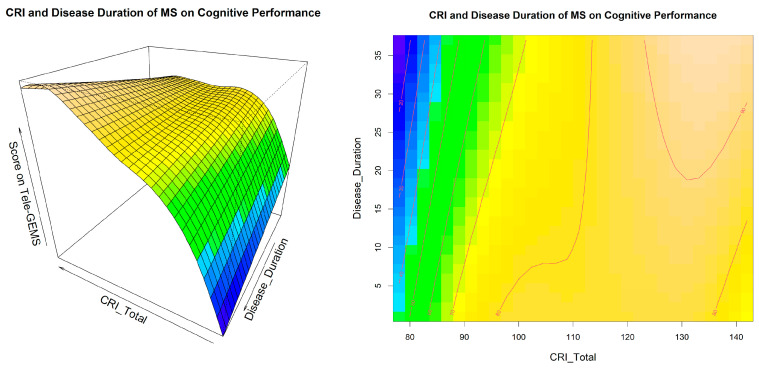
Interaction effect of CRI and disease duration on Tele-GEMS. The figure shows the effect of the tensor surface associated with the interaction between CRI and disease duration (in three-dimensional mode on the left side and in two-dimensional mode on the right side). CRI is on the *x*-axis, disease duration is on the *y*-axis, and Tele-GEMS is on the *z*-axis in the left side and is represented by the red lines on the right side. The darker blue in the contour plots shows a lower Tele-GEMS, whereas darker shades of yellow show a higher Tele-GEMS.

**Figure 2 behavsci-13-00708-f002:**
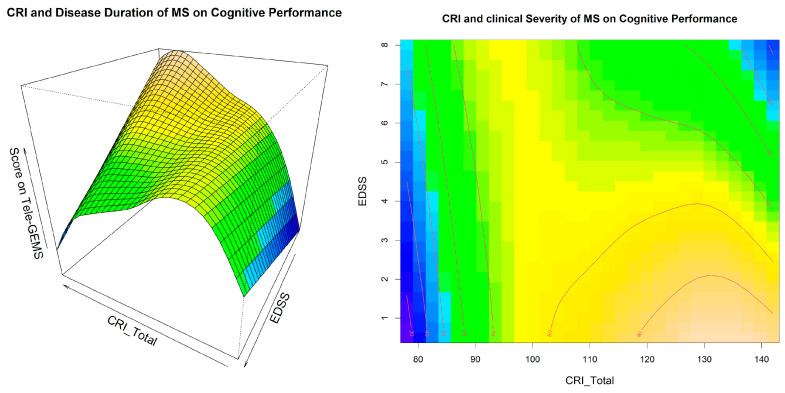
Interaction effect of CRI and EDSS on Tele-GEMS. The figure shows the effect of the tensor surface associated with the interaction between CRI and EDSS (in three-dimensional mode on the left side and in two-dimensional mode on the right side). CRI is on the *x*-axis, EDSS is on the *y*-axis, and Tele-GEMS is on the *z*-axis on the left side and is represented by the red lines on the right side. The darker blue in the contour plots shows a lower Tele-GEMS, whereas darker yellow shows a higher Tele-GEMS.

**Table 1 behavsci-13-00708-t001:** Descriptive statistics of the sample in this study.

	Mean	SD	Median	Min	Max	Kurtosis	Skewness	Q1	Q3
Age	50.89	9.25	51	30	74	0.09	0.04	46	56
Education	12.53	3.53	13	5	21	−0.65	0.04	8	13
Disease Duration	14.82	9.28	13	1	37	−0.75	0.45	7	21
EDSS	3.04	1.90	2.50	1	8	−0.31	0.86	1.5	4
CRI	105.67	13.58	104	78	142	−0.29	0.33	96	115
Tele-GEMS	76.74	15.65	81.8	19.5	98.3	1.78	−1.31	60.08	87.73

Note. EDSS = Expanded Disability Status Scale [12]; CRI = Cognitive Reserve Index [25].

**Table 2 behavsci-13-00708-t002:** Correlation matrix between the variables of interest.

Variable	1	2	3	4	5
1. Age	-				
2. Disease Duration	0.34	-			
3. EDSS	0.30	0.36	-		
4. CRI	0.13	0.05	−0.36	-	
5. Tele-GEMS	−0.29	−0.23	−0.44	0.50	-

Note. EDSS = Expanded Disability Status Scale [12]; CRI = Cognitive Reserve Index [25].

**Table 3 behavsci-13-00708-t003:** Results of generalised additive models with Tele-GEMS as the dependent variable and age, CRI, disease duration, and EDSS as covariates, with CRI and disease duration and CRI and EDSS in interaction.

Generalised Additive Models: Interaction between Cognitive Reserve, Disease Duration, and EDSS											
Tele-GEMS = Dependent Variable											
Model	Intercept			Smooth Terms				Model Fit			
	Estimate (Standard Error)	*t*	*p*	Terms	Edf	F	*p*	R^2^	Dev	AIC	Model Weights
1	76.74 (1.19)	64.71	<0.001	s(Age)	1	3.49	0.065	0.43	46.50%	787.96	0.1
				s(CRI)	3.87	8.56	<0.001				
				s(DD)	1	1.54	0.217				
				s(EDSS)	1	1.82	0.18				
2	76.67 (1.18)	64.71	<0.001	s(Age)	1	2.71	0.103	0.44	49.50%	786.86	0.183
				s(CRI)	3.94	7.67	<0.001				
				s(DD)	1	1.64	0.203				
				s(EDSS)	1.29	0.74	0.337				
				ti(CRI,DD)	1.93	1.41	0.257				
3	76.13 (1.32)	57.59	<0.001	s(Age)	1	3.71	0.567	0.43	47.60%	788.43	0.079
				s(CRI)	4.13	8.13	<0.001				
				s(DD)	1	1.94	0.167				
				s(EDSS)	1	2.54	0.114				
				ti(CRI,EDSS)	1	1.05	0.307				
4	75.05 (1.36)	55.03	<0.001	s(Age)	1	3.37	0.07	0.46	50.90%	784.26	0.638
				s(CRI)	4.27	8.55	<0.001				
				s(DD)	1	2.47	0.12				
				s(EDSS)	1	2.79	0.098				
				ti(CRI,DD)	1	5.7	0.019				
				ti(CRI,EDSS)	1	4.98	0.028				

Note. The table shows, for the four GAMs carried out, the result associated with the intercept, the smooth terms, and the model fit. Within the intercept, the table shows the standard error, *t*-value, and *p*-value associated with the standard error. Within the smooth terms, the table reports the names of smooth terms, estimated degrees of freedom (Edf), F-value, and *p*-value associated with the smooth terms. Within the model fit, the table shows the R-squared with the percentage of deviance explained, the AIC, and the model weights which represent the probability of a model being the best among the given set of models.

## Data Availability

The data presented in this study are available on request from the corresponding author under a reasonable request.

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
