# Peer review of "Clinical Profiles in Multiple Sclerosis: Cognitive Reserve and Motor Impairment along Disease Duration"

_behavsci, 2023, doi:10.3390/bs13090708_

Round 1

Reviewer 1 Report

This cross-sectional study is written well however few things can be considered

Does it have any risk of confounding bias from other factors affecting memory and cognition? please describe that too

The discussion section needs to better explain your findings rather than just comparing them with other studies, like one with motor impairment.

Author Response

Reviewer 1

This cross-sectional study is written well however few things can be considered

  1. Does it have any risk of confounding bias from other factors affecting memory and cognition? please describe that too

We thank the reviewer for this request. First, in our Generalized additive models “age” was always considered in the analyses, being a known variable known as affecting memory and cognition. Second, in the manuscript last paragraph of Discussion section (pag 8) we, indeed, mentioned these possible interactions with confounding factors. Now this part is clearly highlighted: “However, other important variables were possibly at play in affecting cognitive efficiency in our sample such as perceived fatigue, type of medication, mood deflection and psychological traits. These confounding factors have not been considered in the present study, and thus further investigation will be necessary to examine the interaction between CR and cognitive MS symptoms.” 

  1. The discussion section needs to better explain your findings rather than just comparing them with other studies, like one with motor impairment.

We thank the reviewer for this comment. We rearranged in the new manuscript different parts of the Discussion deleting what was not necessary. 

Reviewer 2 Report

The manuscript is consistent throughout, well formulated, and clear in every section. The background is straightforward and complete, the sections on materials and methods are consistent with the results and discussions, and the conclusions are exactly as expected. It has no critical structural, functional, or ethical issues. The abstract has text formatting errors (e.g., interac-tion or neuropsychologi-cal). There are no additional comments to enrich it. 

Author Response

Reviewer 2

The manuscript is consistent throughout, well formulated, and clear in every section. The background is straightforward and complete, the sections on materials and methods are consistent with the results and discussions, and the conclusions are exactly as expected. It has no critical structural, functional, or ethical issues. 

  1. The abstract has text formatting errors (e.g., interac-tion or neuropsychologi-cal). 

We checked all formatting errors, but they were probably due to the template of Behavioral Science.

  1. There are no additional comments to enrich it. 

We thank the reviewer for this appreciation of the study.

Reviewer 3 Report

Dr. Maffezzini's study is based on a telephone assessment of the cognitive performance of people with Multiple Sclerosis, correlating clinical features and cognitive reserve with the subjects' performance.

The study is well written and the results well presented. The main limitation of the work is represented by its limited novelty and impact, given the existing literature on the subject. Furthermore, the cognitive assessment is carried out by means of a screening instrument submitted by telephone, and is therefore limited and only rough (this factor should be discussed as a further limitation of the work). It should be specified in the methods how much, if any, time had elapsed between the last outpatient assessment (from which the patient's edss is deduced) and the performance of the telephone interview. In addition, was the possible presence of clinical/radiological reactivations of disease in the proximity of the telephone interview assessed? This could influence the subject's cognitive performance as described in the literature.

I would suggest implementing the analysis by assessing the possible influence between cognitive reserve and EDSS, correcting for age and disease duration.

The discussion could be expanded by discussing the pathogenetic mechanisms of cognitive impairment during MS, as well as the mechanisms by which increased cognitive reserve may limit/slow down this disease manifestation (dendritic spines, synaptic plasticity, compensation mechanisms, etc.), in order to facilitate the understanding for the reader who is not an MS specialist.

Author Response

Reviewer 3

Dr. Maffezzini's study is based on a telephone assessment of the cognitive performance of people with Multiple Sclerosis, correlating clinical features and cognitive reserve with the subjects' performance.

The study is well written and the results well presented. 

  1. The main limitation of the work is represented by its limited novelty and impact, given the existing literature on the subject. 

We thank the reviewer for this comment which allowed us to better explain the point. We mentioned in the text all previous research about this same topic, and we think the novelty of our study is at least in four main points: 1) the methodological approach adopted, which allowed us to take into account the Disease Duration and the motor impairment (EDSS) in interaction with Cognitive Reserve to predict cognitive outcome. In fact, GAMs allows us to take into account more continuous variable in interaction, in order to disentangle the effective relationship of the independent variables with the dependent one (see pag. 8). 2) The existing literature on the topic, as we specified on pag. 2, is still controversial and in particular 3) so far CR has been measured with a heterogeneous series of proxies without standardised measures whereas our study quantified CR with a standardized questionnaire. 4) Eventually,  our results, indicate that high levels of CR are related to improved cognitive performance even in prolonged disease duration and this has a critical impact in clinical settings. These points are highlighted in the manuscript now.

2. Furthermore, the cognitive assessment is carried out by means of a screening instrument submitted by telephone, and is therefore limited and only rough (this factor should be discussed as a further limitation of the work). 

We thank the reviewer for this request, but we think that our tool to evaluate cognition remotely is instead an added value to the study. In fact, Tele-GEMS is a newly standardised instrument for the brief evaluation of cognition which is proved to be reliable compared to the same in-person screening (GEMS), see Montemurro et al., 2023. Furthermore, the assessment was conducted by telephone by an expert psychologist. Remote assessment is nowadays used and promoted, especially in research with repeated follow-ups, at least when reliable tools are adopted (e.g., Bilder et al., 2020). 

3. It should be specified in the methods how much, if any, time had elapsed between the last outpatient assessment (from which the patient's edss is deduced) and the performance of the telephone interview. 

Thanks for this point which is nowinserted in the manuscript in Procedure: “When a participant was at the Multiple Sclerosis Centre  for diagnosis or clinical follow-ups, in that circumstance, the EDSS was administered, and no more than one week later they were contacted by telephone for cognitive assessment.”

4. In addition, was the possible presence of clinical/radiological reactivations of disease in the proximity of the telephone interview assessed? This could influence the subject's cognitive performance as described in the literature.

We thank the reviewer for this comment. Indeed the participants selected for this study did not have any recent relapses of the disease. This is reported in the new manuscript as an inclusion criteria.  

The radiological reactivation of the disease was not proven during the call, but the clinical evaluation carried out only very few days earlier excluded this possibility. We specified this point in the limitation section of the manuscript (pag. 8).

5. I would suggest implementing the analysis by assessing the possible influence between cognitive reserve and EDSS, correcting for age and disease duration.

We thank the reviewer for this request. We rephrased this part in the manuscript pag. 5 in the Results section and pag. 8 in the Discussion section.

6. The discussion could be expanded by discussing the pathogenetic mechanisms of cognitive impairment during MS, as well as the mechanisms by which increased cognitive reserve may limit/slow down this disease manifestation (dendritic spines, synaptic plasticity, compensation mechanisms, etc.), in order to facilitate the understanding for the reader who is not an MS specialist.

We thank the reviewer for this request. We added this part in the Discussion of the manuscript (pag. 8): This positive effect of CR on cognition could be explained in terms of brain reserve (including the richness of networks and dendritic spines, synaptic density, compensation mechanisms and brain plasticity). All these mechanisms support and maintain better cognitive efficiency [19].

Round 2

Reviewer 3 Report

The Authors addressed all my previous observations.